# New Frontiers in Neurodegeneration and Regeneration Associated with Brain-Derived Neurotrophic Factor and the rs6265 Single Nucleotide Polymorphism

**DOI:** 10.3390/ijms23148011

**Published:** 2022-07-20

**Authors:** Carlye A. Szarowicz, Kathy Steece-Collier, Margaret E. Caulfield

**Affiliations:** 1Department of Translational Neuroscience, College of Human Medicine, Michigan State University, Grand Rapids, MI 49503, USA; szarowi4@msu.edu (C.A.S.); collie68@msu.edu (K.S.-C.); 2Department of Pharmacology and Toxicology, Michigan State University, East Lansing, MI 48824, USA

**Keywords:** BDNF, neurodegeneration, SNP, rs6265, synaptic plasticity, Parkinson’s disease, depression

## Abstract

Brain-derived neurotrophic factor is an extensively studied neurotrophin implicated in the pathology of multiple neurodegenerative and psychiatric disorders including, but not limited to, Parkinson’s disease, Alzheimer’s disease, Huntington’s disease, traumatic brain injury, major de-pressive disorder, and schizophrenia. Here we provide a brief summary of current knowledge on the role of BDNF and the common human single nucleotide polymorphism, rs6265, in driving the pathogenesis and rehabilitation in these disorders, as well as the status of BDNF-targeted therapies. A common trend has emerged correlating low BDNF levels, either detected within the central nervous system or peripherally, to disease states, suggesting that BDNF replacement therapies may hold clinical promise. In addition, we introduce evidence for a distinct role of the BDNF pro-peptide as a biologically active ligand and the need for continuing studies on its neurological function outside of that as a molecular chaperone. Finally, we highlight the latest research describing the role of rs6265 expression in mechanisms of neurodegeneration as well as paradoxical advances in the understanding of this genetic variant in neuroregeneration. All of this is discussed in the context of personalized medicine, acknowledging there is no “one size fits all” therapy for neurodegenerative or psychiatric disorders and that continued study of the multiple BDNF isoforms and genetic variants represents an avenue for discovery ripe with therapeutic potential.

## 1. Introduction

Brain-derived neurotrophic factor (BDNF) is a neurotrophin that functions to regulate and promote neuronal survival, differentiation, and outgrowth of central and peripheral neurons. Other members of the mammalian neurotrophin family include nerve growth factor (NGF), neurotrophin 3 (NT-3), and neurotrophin 4/5 (NT-4/5), and they share more than a 50% sequence homology in their primary structure with BDNF [1]. NGF was the first neurotrophin to be discovered by Rita Levi-Montalcini and Viktor Hamburger in the 1950s [2,3]. Using chick embryos, their work described the observation that neurons die when they lack their targets; research which led to their later revelation that the target was a critical source of a diffusible growth factor eventually identified as NGF [2,3]. In 1982, a few decades following this discovery, BDNF was isolated by Yves-Alain Barde and Hans Thoenen from pig brain [4]. Their research demonstrated that this novel growth factor could induce neuronal outgrowth and survival of cultured embryonic chick sensory neurons [4], supporting the “neurotrophic hypothesis” developed by Levi-Montalcini and Hamburger [2]. Although BDNF had a similar molecular weight to NGF, its functional capacities were distinct, and NGF neutralizing antibodies were not able to block its survival-promoting activity [4]. Follow-up cloning experiments established the identity of BDNF with a unique sequence and structure [5].

Nearly all brain regions have been reported to contain BDNF at varying concentrations, but its specific function depends on stage of development as well as the composition of neuronal, glial, and vascular constituents in the anatomical region [6]. BDNF is abundant in the cortex, hippocampus, and visual cortex. It is also found in the striatum, the substantia nigra, and ventral tegmental areas, though BDNF found in the striatum is supplied by cortical and nigral dopamine neuron afferent projections and not the local neurons themselves [7]. This trophic factor is not solely abundant in the central nervous system (CNS) but is also released in appreciable amounts in the peripheral nervous system and by other non-neuronal cells including lymphocytes, microglia, megakaryocytes, endothelial cells, and smooth muscle cells [8]. BDNF production and signaling is critical for a vast array of neurophysiological processes including, but not limited to, neuronal survival, dendritic spine development, synaptogenesis, neurite outgrowth, neuroprotection, long-term potentiation (LTP), and long-term depression (LTD) (for review [6,9,10,11,12]). BDNF has also been found to be a necessary factor in neurogenesis and osteogenesis in human bone both in vitro and in vivo [13,14].

## 2. Brain-Derived Neurotrophic Factor (BDNF)

### 2.1. BDNF Gene Structure and Isoform Processing

The human BDNF gene is located on chromosome 11p13-14 and is composed of multiple noncoding exons and one coding exon. There are 11 exons that can be alternatively spliced to produce a minimum of 17 transcripts, but each transcript generates the same final protein product [15,16,17]. Of the 11 exons, 9 fall within the 5’ region [18]. The BDNF mRNA transcripts that contain exons II and VII are exclusively expressed in the brain, whereas the transcripts containing exons I, IV, and V are expressed in peripheral tissue; exons VI and IX are broadly expressed [14]. BDNF transcription terminates at two polyadenylation sites within exon IX, thus giving rise to two distinct mRNA populations including short (0.35 kb) or long (2.85 kb) 3’ untranslated regions (UTR) [14,18,19]. These two distinct populations have differing localizations: short UTR BDNF (exon I and IV) transcripts are found in the cell soma, whereas long UTR BDNF transcripts (exon II and IV) are trafficked to dendrites to regulate dendritic morphology and affect LTP [18,20]. 

The major coding sequence of BDNF is present in exon IX at the 3’ end and is translated into an inactive precursor polypeptide (i.e., **preproBDNF**) in the rough endoplasmic reticulum (ER) [8,16,21]. Within the rough ER, the signal sequence is immediately cleaved to yield the 28- to 32-kDa isoform **proBDNF** [8,18] which is comprised of an N-terminal prodomain and a C-terminal mature domain (Figure 1A). Post-translational modifications including N-linked glycosylation of the prodomain as well as sulfation of the N-linked oligosaccharides can take place as the proBDNF neurotrophins migrate from the Golgi apparatus to the trans-Golgi network (TGN). The processing of proBDNF continues via cleavage by intracellular proteolytic enzymes in the TGN (i.e., furin) or by convertases present in intracellular secretory vesicles for extracellular export [22]. A portion of full-length proBDNF proteins is also released and can subsequently bind the high affinity receptor, p75^NTR^ [23]. After release from the cell, extracellular processing of proBDNF by plasmin or matrix metalloproteases (e.g., MMP-2, MMP-9) can also occur (Figure 1B) [8,23,24,25]. Processing of the preproBDNF yields three distinct active isoforms: the ~30kDa proBDNF, the ~13kDa mature BDNF (**mBDNF**), and the ~17kDa BDNF **pro-peptide** [26] (Figure 1).

### 2.2. BDNF Sorting and Release

Two distinct pathways of secretion exist for proBDNF and mBDNF: the constitutive and the regulated pathways. The constitutive pathway involves packaging BDNF into small-diameter granules that release BDNF independently of calcium fluctuation [1]. The majority of BDNF is packaged for release via the regulated pathway into larger granules that fuse to the plasma membrane in response to a calcium-dependent trigger (Figure 1B). Thus, the regulated release of BDNF occurs during activity-dependent depolarization [1,8,28,29] (Figure 1B). Proper sorting and secretion of BDNF is critical for the maintenance of synaptic plasticity, neuronal survival, and CNS homeostasis [1,8,30,31]. As such, disruption of BDNF sorting and/or secretion has been implicated in various neurodegenerative and psychiatric diseases. While the specific molecular mechanisms associated with improper BDNF secretion remain largely uncertain, current evidence correlates reductions of hippocampal and cortical volumes [32], formation of abnormal synapses [33], and decreases in dendritic complexity [34,35] as consequences of dysfunctional BDNF sorting and reduced secretion.

For the regulated pathway, two binding interactions drive sorting of BDNF into vesicles. The BDNF prodomain/pro-peptide region binds directly to either sortilin, a vacuolar protein sorting 10 (Vps10) domain-containing molecule, or carboxypeptidase E [8,18]. Sortilin contains a transmembrane region and a cytoplasmic tail responsible for signaling endosome sorting in the Golgi apparatus [18]. Sortilin and BDNF have been observed to colocalize within large dense-core vesicles, and sortilin truncation mutations result in impaired sorting of BDNF to the regulated pathway, subsequently decreasing activity-dependent release [36]. Similarly, membrane-bound carboxypeptidase E is a glycoprotein that binds BDNF, and knockdown of carboxypeptidase E in mice has also demonstrated a reduction of downstream activity-dependent BDNF release [18,37]. After being sorted into large dense-core vesicles of the regulated pathway, BDNF is generally trafficked to the axon where it can be degraded by the lysosome [38] or secreted into the synaptic cleft in response to neuronal activation where it can activate two classes of receptors, TrkB and p75^NTR^ [39,40,41]. While the majority of BDNF is transported anterogradely, approximately 23% of BDNF is retrogradely transported to dendrites, although the biological significance of its retrograde trafficking has yet to be elucidated [18,42,43].

### 2.3. BDNF Signaling

Neurotrophins are known to bind to two classes of receptors: a tropomyosin receptor kinase (Trk) and a pan neurotrophin receptor (p75^NTR^) which is a member of the tumor necrosis factor super family [44] (Figure 2). More specifically, mBDNF preferentially binds to its high affinity receptor, TrkB, following its release into the synapse [39,40] (Figure 2B). In contrast, proBDNF binds with high affinity to p75^NTR^ [44,45] (Figure 2A). While mBDNF can also bind p75^NTR^, it does so with low affinity [46]. Additionally, the BDNF prodomain/pro-peptide region binds directly to sortilin, thereby participating in proper sorting of this molecule to its regulated pathway [36].

#### 2.3.1. proBDNF and p75^NTR^

ProBDNF binds to p75^NTR^ upon release, stimulating nuclear factor kappa B (NF-κB), c-Jun N-terminal Kinases (JNKs), and Ras homolog family member A (RhoA) signaling that modulate survival, apoptosis, and growth cone motility, respectively [6,44,47,48] (Figure 2A). The specific cascade that is activated is dependent on which receptors are complexed with p75^NTR^. For instance, when complexed with sortilin, pro-apoptotic pathways are activated [45,49]. Recent evidence indicates that signaling through p75^NTR^ can also synergistically aid in TrkB activation [50,51,52]. Specifically, p75^NTR^ can heterodimerize with TrkB, increasing TrkB binding affinity for mBDNF, thus promoting neuronal growth and survival [45,50,52].

#### 2.3.2. mBDNF and TrkB 

Upon mBDNF binding to full-length TrkB, TrkB dimerizes and autophosphorylates several of its tyrosine kinase residues including Y705 and Y706 in the cytoplasmic loop of the kinase domain, as well as Y515 and Y816 [18,53]. Multiple signaling pathways can be triggered once TrkB is activated including the phosphatidylinositol 3-kinase (PI3K), the phospholipase-C-γ1 (PLC-γ1), the guanosine triphosphate hydrolases of RhoA, and the mitogen-activated protein kinase (MAPK)/Ras cascades (reviewed in [44,47,54]). The PI3K pathway engages in pro-survival activity and enhances dendritic growth and branching [55,56]. The MAPK/Ras signaling cascade controls protein synthesis during neuronal differentiation [57]. Lastly, growth of neuronal fibers is activated via activation of RhoA (Figure 2B) [6,44].

**Figure 2 ijms-23-08011-f002:**
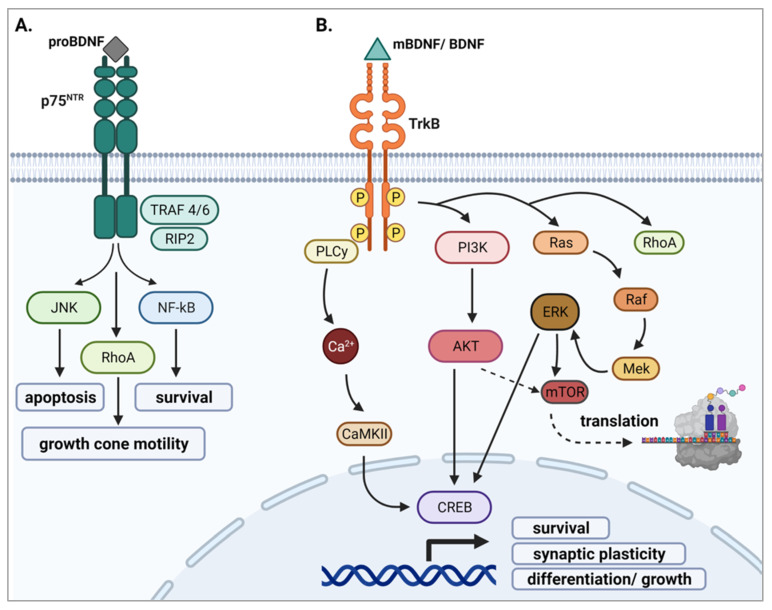
Schematic representations of conventional proBDNF and mBDNF signaling cascades. (**A**) ProBDNF binds with high affinity to p75^NTR^, initiating downstream JNK, RhoA, and NF-kB signaling [6,44,47,48]. (**B**) mBDNF (mature BDNF) binds with high affinity to TrkB, inducing its dimerization and autophosphorylation, thus activating three main signaling pathways, PI3K, PLCγ, and Ras/MAPK, all of which lead to activation of the transcription factor CREB, driving transcription of genes crucial for neuronal growth and survival [44,47,54]. RhoA signaling and mTOR pathways can also be activated leading to growth cone modulation and translation of proteins involved in the regulation of cellular proliferation [18,45,53,56].

It is widely accepted that the proBDNF and mBDNF ligands induce opposing outcomes through their preferential binding to different receptors in order to promote neurological homeostasis [6]. Specifically, mBDNF-TrkB signaling stimulates neuronal growth and synaptic plasticity, whereas signaling through p75^NTR^ tends to initiate apoptosis thought to be important in development for eliminating inessential neurons [48,49]. Moreover, while mBDNF signaling is instrumental in driving hippocampal LTP, proBDNF promotes LTD [58,59,60,61]. Because of homeostatic regulation, the expression of p75^NTR^ and TrkB are known to be tightly linked where they are co-expressed on the surface of the cell to establish signaling between cell survival and cell death [18]. Homeostasis can therefore be disrupted when there is an imbalance in the expression of these receptors or an imbalance in the levels of proBDNF and mBDNF isoforms. For example, research conducted by Suelves and colleagues [62] examined the consequences of BDNF/TrkB/p75^NTR^ imbalance in a Huntington’s disease (HD) mouse model, showing that the reduction of BDNF and TrkB levels, along with an increase in p75^NTR^ expression, correlated with striatal neuropathology and motor dysfunction. Pharmacological normalization of p75^NTR^ levels rescued neuropathology (e.g., dendritic spine density) and motor deficits [62,63]. 

In addition to changes in receptor levels/balance, increased proBDNF levels have been correlated with adverse outcomes in neurodegenerative disorders. Specifically, in mice expressing one BDNF allele with a mutated cleavage site, hippocampal proBDNF levels rose and promoted a decrease in dendritic arborization as well as hippocampal volume [53,61]. Further reinforcing the importance of homeostatic balance in brain health, in Parkinson’s disease (PD), serum levels of proBDNF have been reported to be significantly higher in individuals with early PD as compared to heathy controls, whereas mBDNF levels were significantly lower [64]. Collectively, an abundance of data indicate that tight control of both BDNF ligands and their receptors is critical for proper neuronal function and/or survival.

#### 2.3.3. BDNF Pro-Peptide and Sortilin

It has been demonstrated that BDNF pro-peptide binding to sortilin drives proper sorting of BDNF into vesicles of the regulated secretory pathway [36]. In addition, the BDNF prodomain-peptide (pro-peptide), once cleaved from proBDNF, appears to function as an independent ligand similar to proBDNF and mBDNF isoforms [27,65,66]. Upon cleavage from proBDNF and its subsequent release, the BDNF pro-peptide binds to sortilin and complexes with p75^NTR^, resulting in various effects on the BDNF signaling cascade, neuronal survival, and synaptic plasticity [27,36,65,67,68], although specific mechanisms and downstream pathways remain to be elucidated.

## 3. Genetic Polymorphisms of BDNF

Remarkably, more than one hundred polymorphisms have been described in the BDNF gene [14,69]. While many known variants exist within non-coding regions, understanding of their functional consequences remains limited. However, the most extensively studied single nucleotide polymorphism (SNP) is the Val66Met (G196A, rs6265) polymorphism within the prodomain region of the BDNF gene. Other less well-studied variants exist within this region including Thr2I1e (rs8192466), Gln75His (rs1048221), Arg125Met (rs1048220), and Arg127Leu (rs1048221) and are reviewed elsewhere [14,18,70,71].

### rs6265 (Val66Met)

The rs6265 BDNF SNP, or Val66Met, results from a nucleotide exchange from guanine to adenine at position 196 (G196A). This change results in a substitution of valine to methionine at codon 66, thus referred to as Val66Met [27] (Figure 1C). An individual can be heterozygous (Val66Met) or homozygous (Met66Met) for this SNP. The prevalence of this SNP worldwide is approximately 20% with certain populations in East Asia reporting an incidence up to 72% [33,72,73]. Found in the prodomain region of the BDNF gene, this substitution creates binding interference between the BDNF prodomain/pro-peptide of proBDNF to sortilin. The consequential result, and subsequent hallmark of this polymorphism, is a decrease in activity-dependent release of BDNF, with no reported alterations in constitutive release [14,35]. The reduction in BDNF release is dose-dependent with homozygous subjects showing significantly less release compared to heterozygous subjects (Met/Met > Val/Met > Val/Val) [33].

A wealth of studies have documented a variety of neuropathologies associated with the decrease in secreted mBDNF linked to rs6265 including reduction of hippocampal and cortical volume, abnormal synaptic connections, and decreased dendritic complexity and arborization [20,32,33,34,35,74]. The functional consequences of this common genetic variant are wide-reaching and have been documented to impact memory and cognition, anxiety, and depression, and have been associated with obsessive compulsive disorder (OCD), attention deficit hyperactivity disorder (ADHD), schizophrenia, multiple sclerosis, blepharospasm, and migraine [32,34,35,74,75,76,77,78,79]. Such pathology may be linked to evidence demonstrating that the BDNF Val66Met substitution can result in binding disruption of the translin/trax complex to BDNF mRNA transcripts, subsequently compromising transport of transcripts to dendrites which is critical for synaptic plasticity and dendritic complexity [18,19,20]. As a consequence, decreased BDNF trafficking to dendrites may have negative implications in multiple neurodevelopmental and neurological disorders [20,76]. 

In addition, decreased BDNF levels/signaling have been implicated in neurodegenerative disorders including Alzheimer’s disease (AD), Parkinson’s disease (PD), and Huntington’s disease (HD) (Figure 3A). While it is beyond the scope of this review to provide an extensive analysis of what is known about rs6265 in the aforementioned brain maladies, we briefly highlight how the expression of this common human genetic variant impacts PD, AD, HD, major depressive disorder (MDD), and schizophrenia in the following paragraphs.

## 4. BDNF and the rs6265 SNP in Neurodegeneration

### 4.1. Parkinson’s Disease (PD) and BDNF

PD is a relentlessly progressive neurodegenerative disorder with typical motor symptoms that include rigidity, tremor, and akinesia/bradykinesia, in addition to a host of non-motor symptoms. Motor symptoms are caused by progressive and selective degeneration of dopamine (DA) neurons in the substantia nigra pars compacta (SNpc) and the accompanying loss of afferent nigrostriatal DA innervation. With the exception of monogenic forms of PD, the etiology of idiopathic PD remains unknown (for review [119]).

While dysfunction in BDNF signaling is not considered a primary cause of PD, it has long been known to be important for survival and development of SNpc DA neurons [120,121]. In addition, there is abundant literature demonstrating that, in the aged brain, there is diminished BDNF, diminished upregulation in response to stress, reduced expression of several BDNF transcription factors, and decreased expression of its TrkB receptor (for review [122]). Given that the primary risk factor for PD is aging, and given the critical role of BDNF in the well-being of SNpc DA neurons, BDNF dysfunction has been abundantly explored in PD.

Current evidence has demonstrated reduced expression of BDNF mRNA transcripts in the SNpc in PD [84,123] as well as lower levels of BDNF protein specifically in the SN of individuals with PD compared to other brain regions, and significantly reduced serum BDNF [81]. In addition to decreases in BDNF transcript levels, Scalzo and colleagues [81] have demonstrated that decreased BDNF levels are also detectible in serum of individuals with PD compared to healthy individuals and that concentrations were correlated with PD symptom severity [81] (Figure 3A). However, as the disease progresses, BDNF levels have been shown to increase [81,90,124], thought to be a compensatory mechanism in later diseased states. 

In addition to changes in BDNF in PD, expression of TrkB receptors, which have high expression in SNpc neurons [125], has been shown to be altered in individuals with PD with evidence of isoform-specific alterations. For instance, levels of truncated TrkB have been reported to decrease in axons of the striatum, whereas levels were reported to increase in the striatal soma and distal dendrites of the SN in individuals with PD [126]. Full-length TrkB levels, in contrast, were found to be decreased in striatal neurites and in the cell soma of dendrites, but levels were higher in cell somas and axons of the striatum and SNpc, respectively [126,127]

These findings are corroborated in mouse models of PD where reducing levels of BDNF protein in the SNpc results in a reduction in DA neurons as well as a subsequent decrease in striatal dopamine [83,128]. Further, haplo insufficiency of the BDNF receptor, TrkB, in transgenic mice has been associated with degeneration of SNpc DA neurons over time and in association with aging [122].

#### rs6265 in PD

Although expression of the BDNF rs6265 Met allele is not correlated with an increased incidence of PD, it may contribute to worsening non-motor symptomology [71,129,130]. For example, apathy is one of the most common non-motor neuropsychiatric symptoms of PD [130], and although not statistically significant, PD individuals who were homozygous for the Met allele (Met/Met) were reportedly more likely to display apathetic emotions compared to those without the Met/Met genotype. Moreover, the risks of impulsive-compulsive and related behavioral disorders are also statistically correlated in individuals with PD when expressing the rs6265 SNP [129]. 

An important distinction of Met allele carriers with PD has been in their response to certain pharmacotherapies including levodopa treatment, the gold-standard symptomatic therapy for PD. Specifically, it has recently been reported that Met allele carriers, homozygous or heterozygous, reported worse United Parkinson’s Disease Rating Scale (UPDRS) scores when administered levodopa monotherapy compared to their homozygous Val allele carrier counterparts [102,131]. Individuals expressing the Met allele were also found to have a higher risk of developing the often debilitating side-effect known as levodopa-induced dyskinesia (LID) earlier in their treatment compared to homozygous Val allele carriers [104,132]. 

To contrast these negative correlations of the Met allele, in unmedicated PD patients, a lower severity of motor symptoms has been observed in the initial stages of the disease in BDNF variant individuals [103]. Although homozygous Met allele carriers tended to have more tremor-like symptoms, the progression of the disease was slower, with delayed need for levodopa administration compared to Val allele carriers [103]. Along with this notable decrease in severity of motor symptoms, a later age of onset of PD was reported in homozygous Met allele individuals compared to their Val/Val and Val/Met counterparts with one cohort reporting a 5.3-year later age of onset [105,133] (Figure 3B). In contrast, Svetel et al., 2013 reported that the presence of the Met allele was not associated with clinical characteristics of PD including age of onset and disease severity [134].

This dichotomy of data on the role of rs6265 in PD points to the need for continuing research in this area. In addition, it highlights that there is a significant advantage to tailoring therapeutics in neurodegenerative diseases with the goal of increasing efficacy. Indeed, the inherent heterogeneity in response to therapeutics in individuals with PD [102,131,135] is a significant impediment to implementation of the overall goal of “delivering the right treatment to the right person at the right time to implement safe, effective, and precise interventions with minimal complications” [136]. One approach to deconstructing the complexity of PD and response to therapy is the identification of common genetic variants, like rs6265, that influence these variables as a means to predict disease characteristics and tailor treatments to those most effective for subpopulations of patients.

### 4.2. Alzheimer’s Disease (AD) and BDNF

AD is the leading cause of dementia characterized by severe cognitive decline with age (e.g., [137]). The two pathological hallmarks of AD are extracellular β-amyloid plaques (APs) and intracellular neurofibrillary tangles (NFTs) [138]. There also is significant neuron loss in the hippocampus, entorhinal cortex, and basal forebrain neurons, eventually affecting large portions of the cortex, areas involved with memory and cognition, and eventually language, reasoning, and social behaviors [138,139]. Based on the importance of BDNF in neuron viability, synaptic plasticity, and learning and memory [30,140], this neurotrophin has been of great interest in AD.

As recently reviewed, dysfunction in BDNF and TrkB have been found in AD [141]. Nearly three decades ago, Phillips and colleagues [85] reported lower levels of BDNF mRNA transcripts in hippocampal tissue from AD patients compared to age-matched controls [85]. While limited research suggests variable effects of BDNF mRNA and protein levels in the hippocampus of individuals with AD [123,142], consensus supports that there is an overall decrease of BDNF protein and mRNA in the hippocampus, dentate gyrus, and the frontal and parietal cortex [85,86,87,88,127]. Additionally, lower expression levels of TrkB receptors were found in the hippocampus and frontal cortex in individuals with AD [14,143]. Neurons with NFTs, a major pathological hallmark of AD, were also found to have specifically lower intracellular BDNF protein levels [127], and individuals’ serum levels of mBDNF and proBDNF negatively correlated with AD disease severity [14,118,144] (Figure 3A). While downregulation of BDNF and TrkB have been observed in AD, whether AD-specific pathology (i.e., APs and NFTs) is related to these changes has only recently been examined. Specifically, Ginsberg and colleagues [141] have found that, in individuals with AD, downregulation of BDNF is associated with increased APs in the entorhinal cortex, and downregulation of TrkB is associated with increased NFTs in the entorhinal cortex and APs in the hippocampus.

#### rs6265 in AD

Because BDNF is critical for cell survival and synaptic plasticity, especially in the hippocampus, it is not surprising that expression of a genetic variant (rs6265) and decreased release of this neurotrophin have been associated with AD pathology [108]. A meta-analysis conducted by Fukumoto et al., 2010 reported that women who were Met allele carriers were more susceptible to AD, whereas male Met allele carriers were unaffected [73,106]. This work, and additional supporting studies [107,145,146], demonstrate that rs6265 expression may drive a sex-specific susceptibility to cognitive decline in AD for women but not men [107]. Despite the general paradigm suggesting a correlation between the expression of rs6265 and AD susceptibility, not all research groups have been able to establish the same correlation of rs6265 as a susceptibility factor for AD [147,148,149,150]. Thus, the utility of therapeutically targeting BDNF in AD requires additional investigation (Figure 3B). In addition to a potential association with susceptibility, heterozygous and homozygous carriers of the Met allele with AD have a two-fold and three-fold higher risk for AD-related depressive symptoms, respectively [108].

### 4.3. Huntington’s Disease (HD) and BDNF

HD is an autosomal dominant, progressive neurodegenerative disorder in which striatal medium spiny neurons (MSNs) degenerate with eventual shrinkage of the entire brain, resulting in debilitating motor, cognitive, and psychiatric symptoms [151]. Low levels of BDNF have been considered to play a significant role in the pathogenesis of HD. In healthy individuals, corticostriatal neurons innervate the striatum and release BDNF onto MSNs, and successive BDNF activation of TrkB drives MSNs survival [7]. In HD, however, the mutated huntingtin protein results in altered transcription of the BDNF gene, affects the axonal transport and release of BDNF, and alters TrkB and p75^NTR^ receptors, resulting in a decreased corticostriatal release of BDNF and deficits in neuronal survival signaling [151,152]. In postmortem brain tissue from patients with HD, a distinct reduction of BDNF levels in the caudate and putamen was observed as compared to healthy controls [89]. Yet, in the cerebral cortex and hippocampus, normal BDNF levels were found [89,153]. Levels of BDNF mRNA expression were also affected. For instance, in the cortex of 20 subjects with HD, a significant decrease in both BDNF mRNA and protein levels was reported [153]. Furthermore, TrkB mRNA levels were decreased in the caudate and not the cortex. Its truncated isoform and p75^NTR^, however, were increased in the caudate [153]. Multiple research groups have also reported similar findings in animal models of HD [152,153,154]. For example, in mouse models of early symptomatic HD, TrkB receptors were found to be dysfunctional, failing to engage postsynaptic signaling mechanisms, an effect that was rescued by inhibiting p75^NTR^ signaling [155]. Further, in huntingtin-knockout mice, BDNF mRNA levels were reduced in the cerebral cortex due to its physiological dependence on functional huntingtin protein for proper cellular localization [152] (Figure 3A). This evidence suggests that the imbalance of TrkB and p75^NTR^ expression, along with a reduction of BDNF, may compound and exacerbate HD pathogenesis [153].

#### rs6265 in HD

There are currently few published studies that implicate the expression of the rs6265 SNP in HD pathogenesis. Interestingly, individuals with HD who have the heterozygous Val/Met genotype were reported to have a later age of onset when compared to those who were homozygous for the Val or homozygous for the Met genotype [109]. In contrast, four other studies were unable to confirm any relationship between the expression of this polymorphism and the age of onset in individuals with HD [156,157,158,159]. To the best of our knowledge, no additional analyses have been reported correlating rs6265 and HD, and this potential link represents a prospective area of research development (Figure 3B).

## 5. BDNF and rs6265 in Psychiatric Disorders

### 5.1. Major Depressive Disorder (MDD)

MDD is characterized by lack of pleasure in enjoyable activities, cognitive impairment, sleep and dietary issues, abnormal behavior, and suicidal tendencies [92]. Most studies suggest that there is an association between low levels of BDNF and underlying MDD pathophysiology. Meta-analysis in depressed patients has demonstrated that serum levels of BDNF were much lower than in non-depressed control individuals [91,160]. Additionally, in a French population of individuals who were 65 or older, those who had late-life depression also had elevated methylation of the BDNF gene, leading to a reduction in BDNF gene transcription [94]. These data correlate with other findings that suggest levels of BDNF are low in both the serum and the hippocampus of patients with MDD [93,161,162]. Research involving animal models of depression have demonstrated that BDNF serum levels are reduced, and that this is correlated with the duration of the condition, albeit not symptom severity in MDD (reviewed in [92]) (Figure 3A). BDNF levels are also detrimentally implicated in the context of suicide. Specifically, in the hippocampus and prefrontal cortex of teenagers who committed suicide, postmortem tissue analyses showed significant reductions of BDNF mRNA and protein as well as decreased levels of full-length TrkB expression [95,96,160]. These studies indicate that therapeutic BDNF supplementation may have the potential to alleviate the burden of depression, MDD, and suicide. Indeed, antidepressants are thought to elicit their therapeutic effects, in part, by increasing trophic factors in the brain including significant elevations in BDNF (e.g., [163]). Given the fact that individual responses to antidepressants are widely variable, genetic factors may be involved.

#### rs6265 in MDD

An increased risk and severity of depression is common in individuals who are rs6265 Met allele carriers [110,111] (Figure 3B). In postmortem tissue from victims of suicide with the Met allele, there were lower BDNF levels in the anterior cingulate cortex and caudal brainstem compared with non-depressed subjects, although this did not correlate with an increase in suicidal tendency [164]. Further, an increased instance of BDNF gene methylation was reported in heterozygous (Val/Met) older women with anxiety/depression compared to those who were homozygous (Val/Val) [73,165], again implying that there may be sex-specific consequences of Met allele expression. DNA methylation can upregulate or downregulate corresponding gene expressions and modify related phenotypes [165] with rs6265-associated DNA methylation presumably resulting in loss of function in BDNF signaling [164].

### 5.2. Schizophrenia

Affecting approximately 1% of the worldwide population, schizophrenia is a psychiatric disorder characterized by positive symptoms, negative symptoms, and cognitive impairment [166,167] with an onset of symptoms generally in the mid- to late 20s. These symptoms include visual and auditory hallucinations, delusions, depression, self-neglect, confused cognition, and memory impairments. Physiologically, schizophrenia involves the reduction of gamma-aminobutyric acid (GABA) signaling [92,168]; however, glutamate, acetylcholine, serotonin, and DA are also involved in the pathology of schizophrenia [169]. 

Given the essential role of BDNF in neurodevelopment, synapse regulation, and synaptic plasticity, this neurotrophin has been proposed to contribute, in part, to the pathogenesis of schizophrenia [170]. Most meta-analyses concur that serum levels of BDNF in unmedicated and medicated individuals with schizophrenia are reduced and that BDNF levels decrease further with age [92,171,172]. In the brain of schizophrenia individuals, the dorsolateral prefrontal cortex was shown to contain significantly lower BDNF protein and mRNA, along with decreased TrkB expression, compared to healthy controls [97,98,99,100,101] (Figure 3A). The DNA methylation status of the BDNF gene was reportedly altered in a prenatal stress mouse model of schizophrenia with a consequent decrease in BDNF mRNA and enrichment of 5-methylcytosine and 5-hydroxymethylcytosine at BDNF gene regulatory regions [173]. Increased inflammatory factors such as proinflammatory cytokines have also been reported to be present in postmortem brain tissue from the dorsolateral prefrontal cortex of individuals with schizophrenia, suggesting there may be interplay between DNA methylation, decreased BDNF production, and inflammation in these individuals [92,174]. One contradictory analysis in post-mortem schizophrenic brains demonstrated that BDNF levels were elevated in the hippocampus and anterior cingulate cortex, while the expression level of TrkB was decreased in the hippocampus and prefrontal cortex [175]. Although a correlation between low BDNF levels and/or altered gene regulation and schizophrenia is generally supported in the literature, further studies are necessary to establish the specific mechanisms by which these alterations contribute to schizophrenic pathology and the therapeutic utility of BDNF. 

#### rs6265 in Schizophrenia

Currently, conflicting evidence exists regarding the relationship of rs6265 SNP expression and its alterations in BDNF levels and schizophrenia (for review [176]). In at least two studies, no significant effect of rs65265 genotype profiles were observed for levels of BDNF in serum [166,177]; however, one report found serum levels of BDNF were reduced significantly in individuals with schizophrenia [166], and the other found elevated serum BDNF in schizophrenic subjects [177] compared to control subjects. Indeed, one meta-analysis of approximately 39 case-control studies established that the homozygous Met genotype increased the risk of schizophrenia in Asian, European, and Chinese populations [113]. More specifically, homozygous Met allele carriers had an increased risk of schizophrenia by approximately 19% when compared to Val/Met carriers [113,114]. In contrast, several studies have failed to find an association of rs6265 and schizophrenia (for review [176]). There is, however, some evidence to suggest an association between rs6265 and age of onset in schizophrenia patients with Val/Met males having an earlier age of onset; this association was not noted in females [115,116,117]. It was observed that, in a Caucasian (Armenian) population of chronic schizophrenia individuals, homozygous Met allele carriers, regardless of sex, had an earlier age of onset compared to the individuals who carried Val/Val and Val/Met genotypes [178]. Interestingly, the Val/Val genotype predisposed patients to symptoms that were more severe, exhibiting more clinical symptoms compared to Met allele carriers [117,179] (Figure 3B). Based on the ambiguous nature of current published meta-analyses, additional studies are warranted and should include a focus on additional BDNF polymorphisms as well as sex, age, and ethnic variations.

## 6. Targeting BDNF in the Brain: Therapeutic Challenges and Potential

Overall, BDNF levels are negatively correlated in neurodegenerative and psychiatric disorders. Therefore, many BDNF-targeted therapies aim to raise the levels of BDNF either exogenously or endogenously. Exogenous application of BDNF through direct infusion has been demonstrated to be beneficial to varying degrees in numerous animal studies [180,181,182,183]. As a neuroprotective agent in PD models against DA neuron toxins such as 6-hyroxydopamine (6-OHDA) or 1-methyl-4-phenyl-1,2,3,6-tetrahydropyridine (MPTP), BDNF is effective at protecting SH-SY5Y neuroblastoma neurons in vitro and can modestly protect against 6-OHDA in vivo [183]. Additionally, in an AD rat model, exogenous BDNF application showed dose-dependent neuroprotection against amyloid-β-induced neurotoxicity in cortical neurons in vitro and, when administered intracerebrally, protected cholinergic neurons in the forebrain and diminished the concentrations of amyloid-β peptides [181,182]. Despite promising outcomes from select research conducted in preclinical animal models, a large-scale clinical trial involving oral BDNF supplementation in patients with amyotrophic lateral sclerosis (ALS) did not significantly increase patient survival at dosages of 50–100 mg/day [184]. In a clinical trial involving intrathecal delivery of BDNF to ALS patients, doses of 150 mg/day were well tolerated; however, conclusions about treatment efficacy were unable to be drawn due to small sample sizes [185]. However, a later trial also using intrathecal BDNF for ALS found a lack of clinical efficacy [186]. These disappointing clinical trial results could, in part, be due to the poor pharmacokinetics of BDNF. 

The pharmacokinetics of neurotrophins are complex, making BDNF administration for brain therapeutics especially difficult. Neurotrophins are large, sticky molecules that cannot readily cross the blood-brain-barrier, have short half-lives reported to be 30 min or less [187], inefficiently diffuse into tissues [118], and approaches like intrathecal delivery result in broad exposure to nontargeted structures, thus limiting their scope of effectiveness [118]. If pharmacokinetic barriers could be overcome, consideration needs to be given to therapeutic concentrations of BDNF intended for delivery as well as the availability and status of TrkB receptors. Specifically, exogenous administration of BDNF in regions with significant reductions in TrkB expression, which is known to occur in PD and AD, could severely limit therapeutic benefit. In addition, excessive levels of BDNF could also have a negative impact. Not only can higher concentrations of BDNF downregulate TrkB expression, but excessive amounts of BDNF can lead to unwanted side effects such as seizures, fever, weight loss, fatigue, and diarrhea [127]. Molecularly, excess BDNF can likewise have a negative effect on synaptic circuitry, learning, and memory by inducing hyper-excitation in regions such as the hippocampus [188]. Keeping the above challenges in mind, non-pharmacological methods of BDNF delivery bear potential. 

### 6.1. Gene- and Cell-Based Therapy

A promising non-pharmacological therapeutic technique is in vivo BDNF gene de-livery. This technique involves utilizing viral vectors to transduce host cells with the BDNF gene for downstream endogenous in situ mRNA and protein production. In this way, the high concentrations of local BDNF production in specific regions will ideally protect degenerating neurons in diseases such as PD, HD, and AD [189]. Preclinically, in a post-stroke depression rat model, intranasal delivery of a BDNF-encoding adeno-associated viral vector (AAV-BDNF) increased BDNF mRNA and protein in the prefrontal cortex, alleviating depressive-like symptoms [190]. Additionally, preventative intrastriatal injections of AAV-BDNF reduced the loss of NeuN, a pan neuronal maker, in a lesioned rat model of HD, therefore providing neural protection [191]. Although clinical trials of gene therapy that intended to supplement another neurotrophic factor (i.e., GDNF or neurturin) for neuroprotection against PD have been conducted, results are not yet promising [192,193,194]. Moreover, it remains unknown if it is clinically viable to target low BDNF levels in neurodegenerative or psychiatric disorders via gene therapy. 

Another available BDNF-targeting gene therapy involves an ex vivo autologous approach for neuroregeneration. Briefly, cells such as fibroblasts are taken from the subject, genetically modified to produce BDNF, and then transplanted back into the cell donor’s brain. Like in vivo methods, this strategy could allow for the sustained release of BDNF locally in specific brain regions but advantageously would be poised to avoid immune rejection. Levivier et al., 1995 showed that genetically modified fibroblasts were able to prevent degeneration induced by 6-OHDA in a rat model of PD [195]. Likewise, in a quinolinic acid toxin model of HD, rat fibroblasts were genetically engineered to produce BDNF and transplanted back into the rat brain, resulting in the protection of striatal neurons as compared to control animals [191]. Similarly, mesenchymal stem cells (MSCs) genetically altered to overexpress BDNF have been shown to reduce neuropathological and behavioral deficits in rodent models of HD, suggesting that these approaches have considerable potential for clinical use (for review [196]).

In 2005, a clinical trial was conducted using this ex vivo method in the context of AD for the supplementation of NGF. Eight patients with mild AD received implants of autologous fibroblasts that were genetically engineered to produce NGF and were transplanted into the nucleus basalis of Meynert via stereotaxic surgery [197]. In follow-up studies after 22 months, no long-term side effects occurred, and the rate of cognitive decline was improved in these patients [197]. Considering that this ex vivo autologous treatment was well tolerated, and symptom improvement was demonstrated in AD [189,197], additional clinical trials would seem to be warranted involving BDNF-centric modifications.

Gene therapy, whether viral vector-mediated or autologous transplantation of genetically modified cells, holds strong promise but is not without caveats [198,199]. In general, local release of BDNF is difficult to tightly regulate genetically, and as introduced above, overproduction of BDNF can be detrimental to the circuitry of the brain [118,188]. In addition, both approaches involve invasive surgical protocols; however, in the scope of neurosurgery methods that are much more aggressive (e.g., tumor resection), the approach for vector or cell graft delivery is minimally invasive and straightforward. Of additional concern is immune response to viral vectors and the associated products of foreign transgenes [200]. However, as recently reviewed, current efforts and advances in clinical trials have led to advances to circumvent immune obstacles including modifying AAV capsids to evade pre-existing neutralizing antibodies and development of new methods for clearing of antibodies from circulation (for review [200]). With the advent of new DNA modification techniques, it is not beyond the realm of possibilities that novel gene therapy approaches could be applied in the future. In addition, given that ex vivo autologous treatment was well tolerated, and symptom improvement was demonstrated in AD [189,197], this approach remains hopeful to those suffering from neurodegenerative or neurological disorders.

### 6.2. BDNF Mimetics

One of the most promising BDNF-related administration strategies involves the use of BDNF mimetics. These are small molecules designed to mimic the binding loops of BDNF, resulting in the phosphorylation and activation of TrkB and its downstream effectors, AKT and ERK [118,201,202]. The use of small molecules allows for the delivery of controlled dosages with improved pharmacokinetics compared to full-length BDNF. Mimetics have shown improved diffusivity, blood-brain-barrier permeability, and augmented receptor specificity with less promiscuity [201,202,203,204]. These compounds, however, would require repeat dosing and would not be brain region-specific in targeting, potentially trafficking to areas where their engagement is not advantageous [202,205].

Two common BDNF mimetics are 7,8-dihyrodxyflavone (DHF) and GSB-106. 7,8-DHF is a naturally occurring flavonoid responsible for binding and initiating TrkB signaling pathways. 7,8-DHF application has been investigated in many neurodegenerative and neurological disorders including PD and AD [206,207,208]. For example, in a mouse model of AD, cognitive deficits were restored after 7,8-DHF administration [206,207]. In a comprehensive report by Jang and colleagues [208], 7,8-DHF was documented in mice to specifically activate TrkB in the brain, to diminish kainic acid-induced toxicity in the hippocampus, to decrease infarct volumes in a middle cerebral artery occlusion model of stroke, and it was neuroprotective in a MPTP model of Parkinson’s disease [208]. Cognitive deficits were also rescued with 7,8-DHF treatment in a mouse model of Down syndrome [209]. Collectively, these studies support the idea that 7,8-DHF may be a therapeutic mimetic worth implementing in a wide range of disorders. 

Another common mimetic is bis-(N-monosuccinyl-L-seryl-L-lysine) hexameth-ylenediamide, also referred to as GSB-106, and it mimics the interaction between the TrkB receptor and BDNF via loop 4 of BDNF. Like 7,8-DHF, GSB-106 administration elicits neuroprotective properties by preventing apoptosis in SH-SY5Y cells through the suppression of caspase-3 activity [204]. As reviewed in [210], GSB-106 has also been shown to have a variety of TrkB-mediated neuroprotective effects as well as reduce depressive-like symptoms in a mouse model of depression where administration increased locomotor activity and reduced signs of anhedonia [210,211]. Studies focused on these two BDNF mimetics demonstrate that these small molecules represent potentially useful treatment approaches for those with neurodegenerative diseases and cognitive disabilities. Continued preclinical and clinical development are needed so that their therapeutic effects can be optimized to the greatest extent.

### 6.3. Diet and Exercise

Diet and exercise are widely accessible, non-invasive, low-cost treatments that are of interest for neurodegenerative and neurological conditions. Preclinical studies in various animal models confirm that dietary and exercise regimens increase BDNF levels in the brain and improve cognitive and behavioral functions [118,152,212,213,214,215]. For example, Fahnestock and colleagues [215] demonstrated that implementing a diet high in antioxidants in aged dogs increased BDNF transcripts to levels which were comparative to the young dog cohort [215]. Additionally, restricting the diet of 3-month-old male Sprague Dawley rats to an alternate day feeding regimen compared to ad libitum increased BDNF levels in multiple brain regions including the cortex, striatum, and hippocampus [213]. There also is a wealth of data suggesting that exercise provides neuroprotection in multiple animal models of PD [216,217,218,219,220,221,222] with additional indications that it improves motor symptoms and quality of life in individuals with PD [223,224,225]. Studies using heterozygous deletion of BDNF [226] or inhibition of BDNF TrkB receptors [227] demonstrate that BDNF is essential for the beneficial effects of exercise on the neuroprotection of the nigrostriatal DA system in PD rodent models.

In patients with depression, exercise was found to induce significant increases in serum levels of BDNF levels in all assessed participants [228]. After sprint interval training, BDNF levels were increased directly afterward, then returned to baseline within 90 minutes in eight male subjects [229]. A number of genes, including BDNF, are associated with risk for post-traumatic stress disorder (PTSD), and rs6265 is associated with the psychotic symptoms of PTSD (for review [230]). Intriguingly, in combat veterans with PTSD, active exercise reduced methylation of the BDNF gene at specific CpG sites, resulting in normalized gene expression of BDNF as compared to those without active exercise [230]. Although there are many studies reporting that diet and exercise lead to increased BDNF levels [213,215,229], the specific mechanisms responsible have yet to be elucidated. In the context of BDNF as a therapeutic target, understanding and harnessing the benefits of diet and exercise on BDNF function could lead to vital non-invasive treatments geared toward improving not only neurodegenerative or psychiatric conditions but general patient quality of life.

## 7. The BDNF Pro-Peptide: A Functional “Third Ligand”

Neurotrophin pro-peptides functionally assist in the folding of mature neurotrophin proteins, regulating their localization and bioavailability [231], and the BDNF pro-peptide is no exception. Historically, the BDNF pro-peptide was thought to exclusively function as a molecular chaperone, supporting the folding of mBDNF. Recent studies, however, reveal that the BDNF pro-peptide may also act as an independent functional “third” ligand [27,66,232]. The detection of the BDNF pro-peptide within the laboratory or clinic was previously limited, leading to the conclusion that it may be immediately degraded after proteolytic processing [231,233]. However, in 2012, Dieni and collaborators [43] successfully detected the BDNF pro-peptide and found that both mBDNF and the BDNF pro-peptide were present at equimolar ratios in large dense-core vesicles of excitatory presynaptic termini in hippocampal lysates; an abundance tenfold of that of proBDNF [43,231]. From primary mouse hippocampal neurons, Anastasia and colleagues [27] were also able to isolate the BDNF pro-peptide. The pro-peptide was shown to be released in an activity-dependent manner upon depolarization with potassium chloride like that of the other BDNF isoforms [27]. In addition to its interaction with the sortilin/p75^NTR^ complex, the BDNF pro-peptide has been shown to bind specifically to mBDNF with high affinity in intracellular compartments of trafficking vesicles, representing a potential regulatory mechanism of the mBDNF signaling pathways [67,232] (Figure 4A). 

The BDNF pro-peptide functions as a negative regulator of neuronal structure and survival in multiple circumstances. In hippocampal slices, in the absence of mBDNF, in vitro application of the BDNF pro-peptide facilitated LTD by promoting NMDA-induced AMPA receptor internalization only in p75^NTR^-positive cells [31]. In C6 glioma cells, stimulation with the BDNF pro-peptide decreased cell growth and promoted caspase-3 mediated apoptosis [234]. Comparable results were also found in mature rat hippocampal neurons exposed to BDNF pro-peptide in culture which induced a significant caspase-3-dependent reduction of spine density [235] (Figure 4A). All of these effects were seen using the wild-type (Val/Val) BDNF pro-peptide.

Clinically, the levels of BDNF pro-peptide are altered in individuals with neurodegenerative or psychiatric disorders, but the trends are disease and sex-specific. In human cerebral spinal fluid (CSF), BDNF pro-peptide levels were much lower in individuals with MDD compared to controls, and male patients exhibited significantly lower levels than their female counterparts [241]. Comparatively, in individuals with schizophrenia, the ratio of BDNF pro-peptide to total protein level in CSF was also markedly lower in male patients compared to female patients [242]. In contrast, BDNF pro-peptide levels were significantly increased in postmortem hippocampal extracts from patients with AD; a greater than 30-fold increase [243]. These distinct changes in pro-peptide concentrations may lead to its use as a potential biomarker found in blood and CSF, although additional studies are needed to determine the biological consequences of BDNF pro-peptide flux and clinical feasibility of longitudinal monitoring of pro-peptide levels.

There are also prominent anatomical differences in BDNF pro-peptide expression throughout the body which may determine the role it plays in neurodegenerative or psychiatric disease. In a learned helplessness rat model of depression, the levels of the BDNF pro-peptide were significantly higher in the medial prefrontal cortex (mPFC) of susceptible rats compared to controls, but levels were much lower in the nucleus accumbens (NAc) [244,245]. In the postmortem parietal cortex in patients with MDD, schizophrenia, and bipolar disorder, the levels of mBDNF were significantly lower, whereas pro-peptide levels were significantly higher compared to the control group [245]. In contrast, the cerebellum of these groups with psychiatric disorders contained significantly lower levels of BDNF pro-peptide compared to controls [245]. A negative correlation was also observed between levels of mBDNF in the parietal cortex and mBDNF in the liver of these subjects [245]. 

Disease and region-specific alterations in mBDNF and pro-peptide levels imply that these molecules contribute to the pathology of various psychiatric disorders. In imaging studies of individuals with schizophrenia, the parietal cortex exhibited reduced gray matter [246,247]. Similarly, a reduction in gray matter and dendritic spine number was reported in the mPFC of patients with MDD [248,249,250]. Due to the current evidence demonstrating high levels of the pro-peptide in these brain regions, paired with its negative modulatory effect in various in vitro studies, it is reasonable to hypothesize that therapeutic regulation of BDNF pro-peptide expression would be beneficial, although this strategy would be complicated. Inhibiting pro-peptide expression in the mPFC and parietal cortex, for example, could exacerbate low pro-peptide levels in the NAc [244]. Since one of the many proposed mechanisms of neuropathology in MDD, schizophrenia, and bipolar disorder involve DA neuron malfunction, low levels of pro-peptide in the NAc may be a source of dysfunction in some individuals, and worsening this phenomenon with non-specific pro-peptide lowering strategies may increase disease severity. Conversely, increasing pro-peptide levels in the NAc specifically could be therapeutically beneficial but with potential negative consequences in the mPFC and parietal cortex [251]. 

### Harnessing the Neurogenerative Benefits of the BDNF Pro-Peptide and rs6265 SNP 

Because the BDNF rs6265 SNP involves a substitution of valine to methionine at codon 66, which occurs in this prodomain/pro-peptide region of BDNF that binds directly to sortilin, the Val pro-peptide appears to function differently than that of the pro-peptide containing the Met allele, at least in relation to activity-dependent release of BDNF [252]. As expected, in the hippocampus of homozygous mice expressing the Met allele, lower levels of the Met pro-peptide are released compared to Val pro-peptide secretion [27]. The Met pro-peptide also has a slightly different structure that could impact function. Specifically, in a study conducted by Anastasia and colleagues [27], a shift in conformation was found in the Met pro-peptide using nuclear magnetic resonance [27]. A strong preference for the β-strand conformation is typical; however, the Met pro-peptide tends to demonstrate a helical confirmation. This conformational change resulted in high affinity binding of unique residues in the Met pro-peptide to sortilin with a subsequent downregulation of the Rac pathway, a pathway known to play a role in regulating extension of axons and dendrites, neurite branching, axonal navigation, and synapse formation [27,253] (Figure 4B). In contrast, others have shown that stronger binding of the Val pro-peptide to sortilin occurs as compared to the Met pro-peptide in vitro [36].

In a study that involved treating primary mouse hippocampal neurons with Val and Met pro-peptides, administration of the Met pro-peptide induced growth cone retraction in sortilin- and p75^NTR^-positive cells after a period of approximately 20 min [27]. Similarly, infusing the Met pro-peptide into ventral CA1 hippocampal neurons of mice expressing the Val/Val, Val/Met, and Met/Met genotypes resulted in disassembly of dendritic spines and synaptic elimination, both of which were also dependent upon co-expression of sortilin and p75^NTR^ [68]. Transcript levels of Rac1 were additionally affected after Met pro-peptide infusion and were decreased to half of basal levels, contrasting a robust increase in Rac1 expression levels of mBDNF-treated cells and no change in Val/Val pro-peptide-treated cells [68] (Figure 4B).

Although studies support the paradigm that the proBDNF and the BDNF pro-peptides, whether Val- or Met-type, decrease dendritic spine density, induce growth cone retraction, and promote LTD, these studies have only been conducted thus far in limited brain regions (i.e., hippocampus) (see Table A1). The production and signaling via various BDNF isoforms may elicit distinct biological effects based on isoform ratios and anatomical brain regions. Therefore, further research is necessary to fully elucidate the specific roles of both the Val and Met pro-peptide. 

Despite the tenants of BDNF pro-peptide function discussed above, recent reports paradoxically suggest that the expression of the BDNF rs6265 SNP may confer protective, or neuroregenerative, effects in injury or disease. For instance, expression of the Met allele correlates with improved recovery from brain injury and promotion of axon regeneration in combat veterans who were exposed to traumatic brain injury (TBI) [237,238,239]. A reduction in cognitive decline in individuals with multiple sclerosis and risk of late-onset AD in individuals carrying the Met allele have also been reported [240,254,255]. Multiple rodent studies have likewise reported positive contributions of the Met allele on neuroregeneration [26,236]. Murine Met allele carriers showed enhanced peripheral axon regeneration in vivo and neurite outgrowth in culture compared to the homozygous Val allele carriers [26,236]. Moreover, in a rat model of PD, Met/Met host animals demonstrated robust enhancement of neurite outgrowth derived from intrastriatally placed wild-type midbrain DA grafts and significantly enhanced behavioral efficacy compared to Val/Val hosts despite uniform survival of grafted DA neurons between groups [33]. While the functional benefit of the graft was much greater in Met/Met animals, these animals uniquely developed an aberrant side effect known as graft-induced dyskinesias (GID) seen in a subset of human PD graft recipients and under some conditions in parkinsonian rats (for review [33]). This GID behavior was correlated with grafted neurons maintaining an “immature” neurochemical phenotype possibly attributed to decreased BDNF in the environment of the grafted embryonic DA neurons [33]. The increased neurite outgrowth could be speculated to be due to potential growth enhancing properties of the Met pro-peptide per the data discussed above [33] (Figure 4C).

Overall, these findings contradict the paradigm that the Met allele is solely a “risk” allele through its impact of reducing mBDNF release responsible for detrimental effects seen in many neurodegenerative diseases. Instead, accumulating studies suggest that rs6265 may confer differing mechanisms in varying contexts under normal and pathological conditions [236,237,238,239]. It is interesting to speculate that a reduction in activity-dependent release of the pool of BDNF isoforms may be neuroprotective due, for example, to a proportional decrease in release of proBDNF, BDNF pro-peptide, or altered isoform ratios that interact with p75^NTR^ receptors that can initiate apoptosis. It is likely that, with the reduction of activity-dependent release of BDNF in individuals expressing rs6265, there is parallel reduction in the release of the pro-peptide. This reduction could decrease the negative regulation of mBDNF when bound by the BDNF pro-peptide, amplifying neurite and axonal growth. Finally, because the rs6265 SNP is in the BDNF prodomain/pro-peptide, the unexpected benefit of this genetic variation in TBI [237,238,239], stroke [256], and neural grafting [33] suggests that there is likely an important role for the Met pro-peptide in neuroregeneration. Understanding the “good” and the “bad” associated with rs6265 will doubtless aid in the development of safe and optimized therapeutic approaches of neurodegeneration and other brain disorders. Indeed, a growing collective body of evidence suggests a protective role for the rs6265 SNP in various neurological conditions and could explain the significant prevalence of this genetic mutation in the human population.

## 8. Summary

It is well accepted that mBDNF via interactions with TrkB and p75^NTR^ is critical for the development and maintenance of the numerous populations of neurons in the central and peripheral nervous systems with direct effects on neuronal survival, synaptic plasticity, and the development and maturation of neurite outgrowth. Pro-apoptotic pathways activated through proBDNF- p75^NTR^ engagement are also critical for normal brain homeostasis via neuronal elimination of non-essential neurons which is crucial for proper circuitry development and function. To date, preclinical and clinical studies have reported a generalized decrease in serum and brain BDNF levels in multiple neurodegenerative and psychiatric disorders, including PD, AD, HD, MDD, and schizophrenia. However, contradictory results do exist regarding these trends and their pathological consequences. The paradoxical role of the BDNF prodomain/pro-peptide in brain health and disease, especially in individuals with the rs6265 genotype, warrants additional investigation. The heterogeneity in current findings could be attributed to sex differences, environmental exposures, and co-expression of multiple genetic variants or gene-gene interactions that are yet to be appreciated.

This gap in our understanding of BDNF in the context of neurodegeneration and neuroregeneration is an attractive target for therapeutic advancement with individualized medicine. In a precision-medicine-based climate, successful treatment options will not be accomplished by a “one-size fits all” method. In heterogenous disorders such as PD and AD, a patient would more likely benefit from having their individual characteristics such as genetics, age, and sex taken into consideration in treatment plan development. The status of BDNF is no exception. With current BDNF-related research being relatively novel and somewhat contradictory, particularly with regard to the function of proBDNF and the BDNF pro-peptide, continued exploration of BDNF signaling mechanisms in various genetic and environmental contexts is critical for the advancement of BDNF-based therapy. If specific contextual aspects of BDNF and its isoforms can be established, then these mechanisms can be manipulated in a beneficial way to successfully treat patients affected with a range of neurodegenerative and psychiatric disorders.

## Figures and Tables

**Figure 1 ijms-23-08011-f001:**
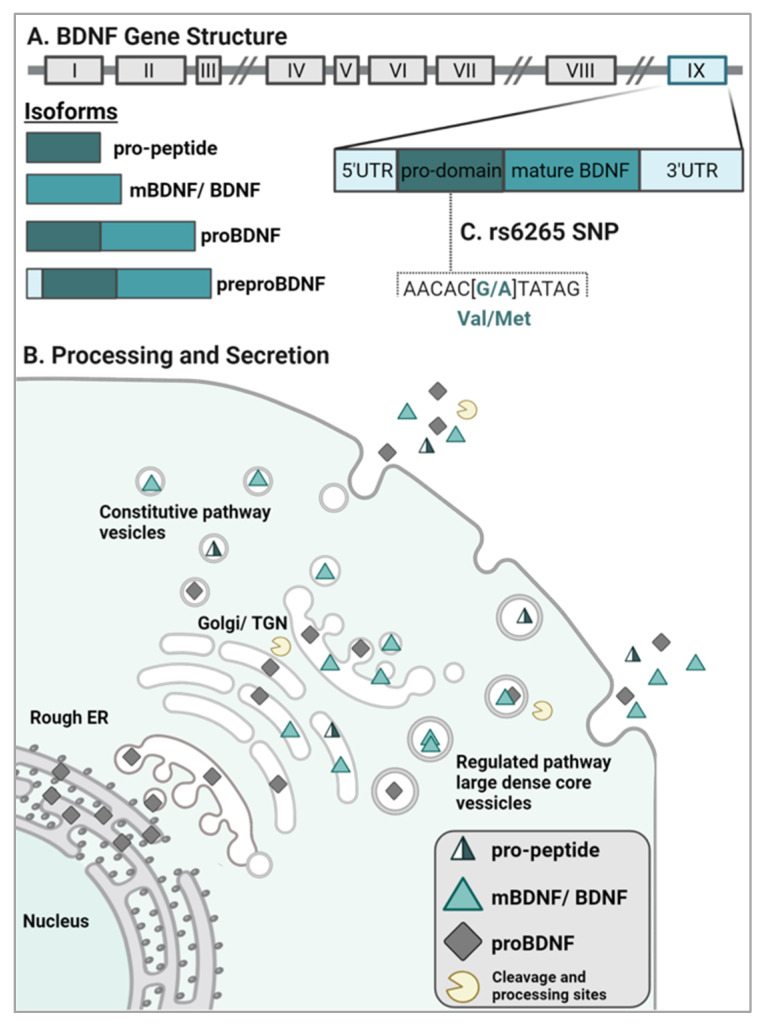
BDNF Gene Structure, Processing, and Secretion. (**A**) Schematic representation of human BDNF gene structure and isoforms. Grey boxes represent exons; exon IX (blue) contains the major coding sequence of BDNF [8,16,21]. (**B**) Following translation into preproBDNF in the ER, the signaling sequence is cleaved, and proBDNF is transported through the Golgi apparatus to the trans-Golgi network. Here, proBDNF can be cleaved by intracellular proteolytic enzymes sorting into the constitutive or regulated pathways [8,22]. ProBDNF can also be cleaved within the vesicles or extracellularly, generating mBDNF and the BDNF pro-peptide [26]. (**C**) The common SNP rs6265 (aka: Val66Met) is located within the prodomain region of the BDNF gene and results in a substitution of valine (Val) for methionine (Met) at codon (G/A) 66 [8,18,27]. Abbreviations: pro-peptide = cleaved BDNF pro-peptide; mBDNF/BDNF = mature BDNF; proBDNF = BDNF isoform with pro-domain and mature domain.

**Figure 3 ijms-23-08011-f003:**
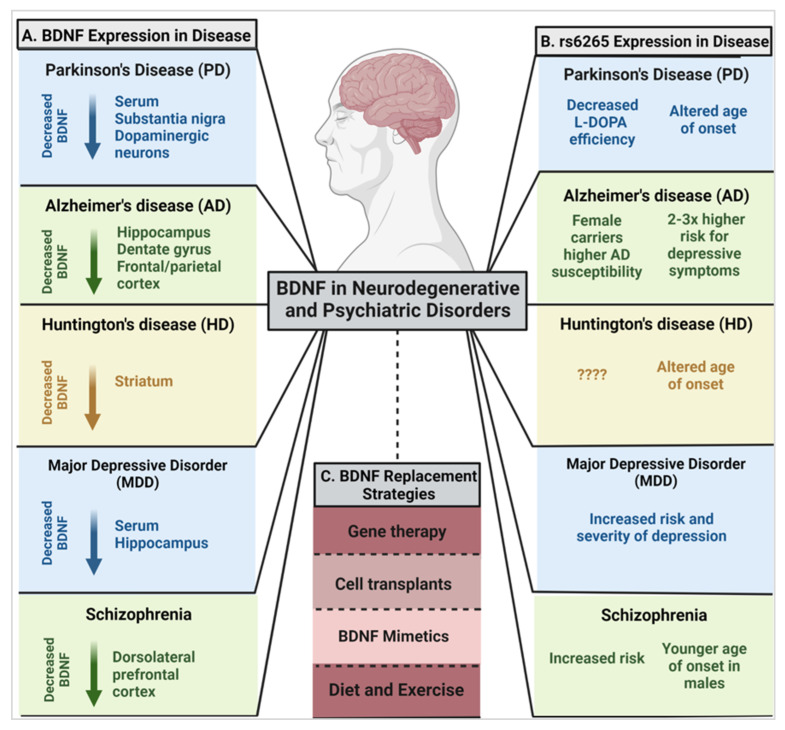
Summary of altered BDNF expression levels and consequences of the rs6265 SNP in neurodegenerative and psychiatric disorders. (**A**) Decreased BDNF mRNA and protein expression in various regions of the brain in PD [80,81,82,83,84], AD [85,86,87,88], HD [89,90], MDD [91,92,93,94,95,96], and schizophrenia [97,98,99,100,101]. (**B**) Associations of rs6265 SNP expression and disease state including therapeutic efficacy, age of onset, and susceptibility to the disease: PD [102,103,104,105], AD [106,107,108], HD [109], MDD [110,111,112], Schizophrenia [113,114,115,116,117]. (**C**) BDNF replacement strategies currently being implemented preclinically and clinically (reviewed in [118]).

**Figure 4 ijms-23-08011-f004:**
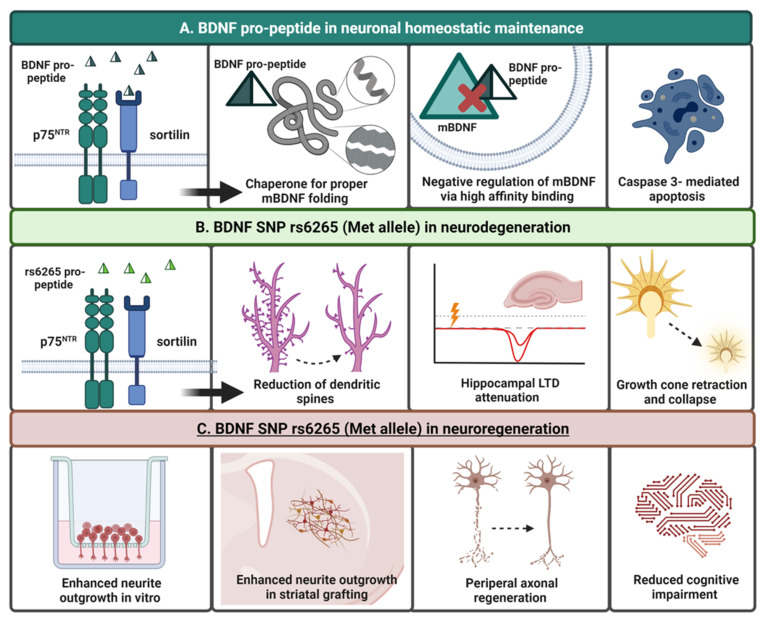
Proposed mechanisms of the BDNF pro-peptide and the rs6265 SNP binding and action in neurodegeneration and neuroregeneration. (**A**) Although it was widely accepted that the BDNF pro-peptide functions as a molecular chaperone, it can also play a role in homeostatic maintenance via binding to or inducing apoptosis through caspase-3 mediate pathways [68,234,235]. (**B**) Application of the Met pro-peptide, and not the Val pro-peptide, reduced dendritic spine density in cultured hippocampal cells [27]. Growth cone retraction was induced by application of Met pro-peptide to hippocampal neurons in culture [27], as was an attenuation of Val pro-peptide-induced hippocampal LTD [31]. (**C**) It is hypothesized that application of the Met pro-peptide may directly enhance neurite outgrowth [33], improve peripheral axonal regeneration [236], and reduce cognitive impairment [237,238]. These hypotheses are built on studies from rodent models of PD as well as studies involving veterans suffering from TBI [237,238,239] and patients with MS [240].

## Data Availability

Not applicable.

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
