# Peer review of "New Frontiers in Neurodegeneration and Regeneration Associated with Brain-Derived Neurotrophic Factor and the rs6265 Single Nucleotide Polymorphism"

_ijms, 2022, doi:10.3390/ijms23148011_

Round 1
Reviewer 1 Report
Nothing
Author Response
Thank you for your review and ranking of this manuscript. Your time and consideration are appreciated.
Maggie Caulfield, PhD
Reviewer 2 Report
The article "New Frontiers in Neurodegeneration and Regeneration Associ ated with Brain-derived Neurotrophic Factor and the rs6265 Single Nucleotide Polymorphism" is well-written review. I have some minor comments that can improve the manuscript.
-Lines 225-226. Why "hallmark of this polymorphism" is underlined?
-Figure 1. What is the mBDNF/BDNF?
-Generally at Figures the abbreviations must be added, so they can be understood/read by themself
-Why Authors did not include other disorders that have been ascoiated with the rs6265 (e.g multiple scleroris, dystonia, blepharospam, headhache...). Maybe they have to discuss the role of the SNP at these disorders at the dicsussion section or to add a limitation/explanation for not including these phenotypes.
-Lines 700-702 "These distinct changes in pro-peptide concentrations may lead to its use as a potential biomarker, although additional studies are needed to determine the biological consequences of BDNF pro-peptide flux and clinical feasibility of longitudinal monitoring of pro-peptide levels." May this important section could be discussed further. CSF or blood or other seems to be the most promising tissue for measuring?
Author Response
-Lines 225-226. Why "hallmark of this polymorphism" is underlined?
Thank you for this comment. The underline formatting of this section was done to draw the reader’s attention to this crucial detail of the underlying mechanistic consequence of the rs6265 polymorphism. The underline was extended to include the following information in and attempt to clarify the formatting. “hallmark of this polymorphism, is a decrease in activity-dependent release of BDNF” (Line 228 -229).
-Figure 1. What is the mBDNF/BDNF?
-Generally at Figures the abbreviations must be added, so they can be understood/read by themselves
Thank you for noticing the lack of description of abbreviation used in Figure 1. Abbreviation descriptions have been added to the figure legend for Figure 1 (Line 102- 104) as well as in the legend for Figure 2 (Line 170).
-Why Authors did not include other disorders that have been ascoiated with the rs6265 (e.g multiple scleroris, dystonia, blepharospam, headhache...). Maybe they have to discuss the role of the SNP at these disorders at the dicsussion section or to add a limitation/explanation for not including these phenotypes.
Thank you for drawing our attention to the need to refer to these other disorders. While our focus of this review is on the role of the rs6265 SNP in neurodegeration, with a focus on Parkinson’s disease, an explanation of the limits of the review and references to other disorders were added (Lines 239-240)
-Lines 700-702 "These distinct changes in pro-peptide concentrations may lead to its use as a potential biomarker, although additional studies are needed to determine the biological consequences of BDNF pro-peptide flux and clinical feasibility of longitudinal monitoring of pro-peptide levels." May this important section could be discussed further. CSF or blood or other seems to be the most promising tissue for measuring?
We agree this concept is important and could be the focus of a larger review. However, with the scope of our review we believe it is important to reference this potential, but an extensive treatment of this topic is outside our focus. Regardless, we mentioned that blood and CSF as probably tissue for biomarker monitoring. (Line 703).